# Individualized Fortification Based on Measured Macronutrient Content of Human Milk Improves Growth and Body Composition in Infants Born Less than 33 Weeks: A Mixed-Cohort Study

**DOI:** 10.3390/nu15061533

**Published:** 2023-03-22

**Authors:** Manuela Cardoso, Daniel Virella, Ana Luísa Papoila, Marta Alves, Israel Macedo, Diana e Silva, Luís Pereira-da-Silva

**Affiliations:** 1Nutrition Unit, Maternidade Dr. Alfredo da Costa, Centro Hospitalar Universitário de Lisboa Central, Centro Clínico Académico de Lisboa, 2890-495 Lisbon, Portugal; 2Research Unit, Centro Hospitalar Universitário de Lisboa Central, Centro Clínico Académico de Lisboa, 1169-045 Lisbon, Portugal; 3Neonatal Intensive Care Unit, Hospital Dona Estefânia, Centro Hospitalar Universitário de Lisboa Central, Centro Clínico Académico de Lisboa, 1169-045 Lisbon, Portugal; 4Centre of Statistics and Its Applications, University of Lisbon, 1749-016 Lisbon, Portugal; 5Neonatal Intensive Care Unit, Maternidade Dr. Alfredo da Costa, Centro Hospitalar Universitário de Lisboa Central, Centro Clínico Académico de Lisboa, 2890-495 Lisbon, Portugal; 6Faculty of Nutrition and Food Sciences, University of Porto, 4150-180 Porto, Portugal; 7CINTESIS—Center for Health Technology and Services Research, 4200-450 Porto, Portugal; 8Nutrition Lab, Hospital Dona Estefânia, Centro Hospitalar Universitário de Lisboa Central, Centro Clínico Académico de Lisboa, 1169-045 Lisbon, Portugal; 9Medicine of Woman, Childhood and Adolescence Academic Area, NOVA Medical School, Universidade Nova de Lisboa, 1349-008 Lisbon, Portugal; 10CHRC—Comprehensive Health Research Centre, Nutrition Group, NOVA Medical School, Universidade Nova de Lisboa, 1349-008 Lisbon, Portugal

**Keywords:** adiposity, body composition, growth velocity, human milk fortification, preterm infants

## Abstract

The optimal method for human milk (HM) fortification has not yet been determined. This study assessed whether fortification relying on measured HM macronutrient content (Miris AB analyzer, Upsala, Sweden) composition is superior to fortification based on assumed HM macronutrient content, to optimize the nutrition support, growth, and body composition in infants born at <33 weeks’ gestation. In a mixed-cohort study, 57 infants fed fortified HM based on its measured content were compared with 58 infants fed fortified HM based on its assumed content, for a median of 28 and 23 exposure days, respectively. The ESPGHAN 2010 guidelines for preterm enteral nutrition were followed. Growth assessment was based on body weight, length, and head circumference Δ z-scores, and the respective growth velocities until discharge. Body composition was assessed using air displacement plethysmography. Fortification based on measured HM content provided significantly higher energy, fat, and carbohydrate intakes, although with a lower protein intake in infants weighing ≥ 1 kg and lower protein-to-energy ratio in infants weighing < 1 kg. Infants fed fortified HM based on its measured content were discharged with significantly better weight gain, length, and head growth. These infants had significantly lower adiposity and greater lean mass near term-equivalent age, despite receiving higher in-hospital energy and fat intakes, with a mean fat intake higher than the maximum recommended and a median protein-to-energy ratio intake (in infants weighing < 1 kg) lower than the minimum recommended.

## 1. Introduction

Mother’s own milk (MOM) is recommended as the first choice for feeding preterm infants and donor human milk (DHM) is the best alternative when MOM is not available [1,2]. Human milk (HM), either MOM or DHM, has biological components that compensate for the immaturity of various organs and systems [3,4,5,6]. However, HM does not fulfill the needs of growing preterm infants from a nutritional point of view, thus resulting in accumulated nutrient deficits, as well as the risks of growth restriction and poor neurodevelopment [7,8]. The main insufficiency of milk from women who give birth to very preterm infants is the protein deficit, which becomes more pronounced as lactation is prolonged [9].

To prevent nutritional deficits in HM-fed preterm infants, while taking advantage of HM biological properties, HM multi-nutrient fortifiers are used, containing protein, carbohydrates, minerals, and vitamins, as well as some fortifiers containing fat [2,10,11]. The more utilized method is the standard fortification that consists of the addition of a fixed dose of an HM multi-nutrient fortifier, according to the manufacturer’s instructions [2,11]. This method overlooks the great variability in the macronutrient content of HM [2,10,11,12], thereby increasing the risks associated with energy–protein malnutrition, [13,14,15]. In an attempt to overcome this problem, individualized HM fortification methods have been proposed, particularly target fortification, which is guided by regular measurements of the energy and macronutrient content of HM, in order to customize the target energy and macronutrient requirements for preterm infants [2,16]. Accordingly, modular protein, carbohydrates, and fat supplements are added to the fortified HM when necessary [2,17,18] to achieve the desirable macronutrient targets that are recommended by the European Society for Paediatric Gastroenterology Hepatology and Nutrition (ESPGHAN) [19]. 

The current literature indicates that very preterm infants at term age have a lower fat-free mass (FFM) and a greater fat mass percentage (%FM) when compared with term-equivalent-age (TEA) infants [20,21,22]. These differences tend to disappear by 52 weeks postmenstrual age (PMA) [22]. Interestingly, it was reported that HM feeding promotes FFM deposition in preterm infants [23,24].

Certain questions remain yet unanswered. First, despite many authors having considered target fortification as a more accurate method to achieve adequate intake of energy and macronutrients in preterm infants [18,25,26], others have reported that this method, when compared with standard fortification, does not meet the recommended protein intake [27], nor does it result in better growth [28,29]. Second, to comply with the ESPGHAN 2010 enteral nutrition guidelines for preterm infants [19], more studies are required in order to determine the maximum safe concentrations of multi-nutrient fortifiers, as well as the modular protein supplements that are added to the limited volume of the milk prescribed, taking into account the osmolality of feeds, feeding tolerance, and the risk of necrotizing enterocolitis (NEC) [28]. Third, only reference values for body composition measurements are currently available for follow-up on very preterm infants [20,21]. Standard prescriptive charts are unavailable due to the difficulty in defining an optimal nutrition in these infants, a major factor that determines body composition [30,31].

In a previous cohort study of preterm infants born at less than 33 weeks’ gestation fed standard fortified HM, we observed that the energy, protein, and fat intakes based on measured HM content were significantly lower than the intakes based on assumed HM content [32]. Further, this nutritional support resulted in a suboptimal weight gain velocity and adiposity deficit [33]. From this evidence, a nutritional strategy for HM fortification based on measured HM macronutrient content was adopted in our unit.

The aim of this study was to assess whether the newly implemented fortification method based on measured HM macronutrient content improves the energy and macronutrient intake and the quality of growth in infants born at less than 33 weeks’ gestation, compared with HM fortification based on assumed HM macronutrient content. Weight gain velocity was the primary outcome. In-hospital length and head growth velocities and body composition at late preterm or TEA were secondary outcomes.

## 2. Methods

### 2.1. Study Design and Ethical Issues

This is a single-center, observational mixed-cohort effectiveness study in preterm infants with less than 33 weeks of gestation [34]. Energy and macronutrient intake, growth, and body composition were compared in infants fed fortified HM, either based on assumed HM macronutrient content (Group 1) or based on measured HM macronutrient content (Group 2). 

Although the original design intended to compare, ‘per protocol’, the effects of standard and target HM fortification methods, as stated in the ClinicalTrials.gov NCT04400396 registry and the published protocol [34], addition of modular protein and fat supplements to fortified HM was left to clinical discretion. As such, comparisons were made ‘per intention to treat’. Overall, the neonatal unit nutritional policy followed the ESPGHAN 2010 enteral nutrition guidelines for preterm infants [19]. Instead of strictly following the standard fortification method by simply adding a fixed dose of the fortifier to the HM, physicians were allowed to further add modular supplements of protein and mid-chain triglycerides (MCT) to the fortified HM, based on assumed HM macronutrient content, according to a reported longitudinal analysis of MOM macronutrient content [35]. Instead of target fortification, the physicians were asked to customize the HM fortification as guided by the measured HM macronutrient content, thereby adding modular supplements of protein and MCT to the fortified HM, if necessary. 

The study was approved by the hospital ethics committee (Nr 558/2018) and informed written consent was obtained from the parents or legal representatives of each infant. 

### 2.2. Settings, Participants, and Study Periods

The study was conducted at the Centro Hospitalar Universitário de Lisboa Central, specifically in the Neonatology Unit and in the Human Milk Bank of Maternidade Dr. Alfredo da Costa, as well as in the Nutrition Laboratory of the Hospital Dona Estefânia.

The eligibility criteria [34] were those used in a historical cohort study [33], with sampling of consecutive infants born with less than 33 weeks of gestation, singletons or twins (=2), appropriate for gestational age, and exclusively or predominantly (>87.5% volume per day) HM-fed. Infants diagnosed with an innate error of metabolism were not recruited.

The exposure period was defined by feeding fortified HM for at least 2 weeks. Recruited infants were dropped out if they were fed formula for two or more days with a >12.5% daily volume intake, or if they were transferred or deceased before completing the exposure period. 

The recruitment for the contemporary cohort of infants fed measurement-based fortified HM was initially planned to start in February 2020 (ClinicalTrials.gov NCT04400396) to compare them with the historical cohort (2014–2015) of infants, who were fed fortified HM based on its assumed macronutrient content [32,33]. However, a shortage in modular protein supplement from 1 February to 13 July 2020 precluded the adoption of the new institutional nutrition protocol, which implied the possible addition of modular protein to the fortified HM guided by the measured HM macronutrient content; meanwhile, fortification continued to be based on assumed HM macronutrient content. Thus, the arm of fortification based on assumed HM macronutrient content included not only the historical cohort of 2014–2015 [33], but also part of the contemporary cohort recruited from 1 February to 13 July 2020 (Table 1). The cohort of fortification based on measured HM macronutrient content was recruited from 14 July 2020 to 31 May 2021 (Table 1).

The required sample size was estimated to be 68 infants, with 34 infants in each group [34].

### 2.3. Institutional Nutrition Protocol

As detailed elsewhere [33,34], the institutional neonatal nutrition protocol was followed considering the more recent international and national guidelines for parenteral and enteral nutrition [19,36,37,38,39]. In brief, parenteral nutrition was initiated within the first 2 postnatal hours and early trophic feeding was initiated within the first 2 to 4 postnatal days preferentially using MOM, and DHM when MOM was insufficient.

Addition of a powdered, multi-component HM fortifier (Aptamil FMS; Danone GmbH, Friedrichsdorf, Germany), containing 3.47 kcal of total energy, 0.25 g of protein, and 0.62 g of carbohydrates per g of powder, was started at a dose of 4.4 g/100 mL HM when the HM intake was at least 80 mL/kg/day. For fortified HM based on its measured macronutrient content, the MOM was analyzed once a week and the DHM was analyzed after pasteurization [34].

The addition of modular protein and fat supplements to fortified HM was prescribed per clinical discretion, whatever the fortification method. According to the HM macronutrient content, a powdered modular protein (Aptamil Protein Supplement; Danone GmbH, Friedrichsdorf, Germany), containing 3.38 kcal of total energy and 0.821 g of protein per g of powder, and/or medium-chain triglycerides (MCT Oil; Danone, GmbH, Friedrichsdorf, Germany), containing a total energy of 8.6 kcal and 0.95 g of fat per 1 mL, were added to the fortified HM, when necessary, in order to follow the ESPGHAN 2010 guidelines and reach the following target daily intakes for actual body weight: 110–135 kcal/kg of total energy, 4.0–4.5 g/kg of protein for infants with <1 kg body weight and 3.5–4.0 g/kg of protein for ≥1 kg body weight, and a protein-to-energy ratio (PER) of 3.6–4.1 for <1 kg body weight and 3.2–3.6 for ≥1 kg body weight [19]. Although the ESPGHAN 2010 guidelines for the intake of protein and PER is stratified by infant body weight <1 kg or 1–1.8 kg [19], we used the convenience criterion of <1 kg and ≥1 kg body weight instead.

### 2.4. Retrieved and Measured Variables

Collected variables: The recorded demographic variables included gestational age at birth, classified as ≥28 weeks or <28 weeks (extreme preterm); sex; singleton or twin; birth weight—small, appropriate, or large for gestational age (<3rd percentile, ≥3rd percentile and ≤97th percentile, and >97th percentile, respectively) [40]; severity index (SNAPPE II) [41]; prenatal corticosteroids; diagnosis of late sepsis [42]; NEC (grade ≥ 3) [43]; intra-periventricular hemorrhage (grade ≥ 3) [44]; multicystic periventricular leukomalacia [45]; and chronic lung disease [46].

Analysis of HM macronutrient content: The HM samples were analyzed for macronutrient content using a real-time HM analyzer (Miris AB, Uppsala, Sweden) [47], following the same measurement procedures described in the historical study [32]. The standard calibration solution was used to calibrate the analyzer before each use [17]. To minimize the daily variability in breast milk macronutrient content, mothers were asked to add milk collected in the last 24 h to the same container. The energy and macronutrient content is expressed in densities; that is, kcal/dL of energy and g/dL of fat, raw and true protein, carbohydrates, and ashes. In the fortification based on the measured HM macronutrient content method, the adjustment of the MOM fortification for the infants’ nutritional needs was guided by a weekly analysis of MOM macronutrient content. The adjustment of the DHM fortification was based on its macronutrient content, which had been previously analyzed and indicated on the containers’ labels.

Assessment of nutrient intake: The daily record of nutritional intake refers to feeds administered in the last 24 h. For each infant, the volumes of LH administered (MOM or DHM) were accurately recorded, considering the amounts actually administered, taking into account possible interruptions and gastric residual volumes. To calculate macronutrient (g/mL) and energy (kcal/mL) densities of fortified HM, the macronutrient content of HM administered was recorded daily, as well as the macronutrient content of the added HM fortifier, modular protein, and MCT oil. The measured HM macronutrient content was used for calculations in the measured HM macronutrient content group and the estimated HM macronutrient content based on literature data [43] was used in the assumed HM macronutrient content group. The macronutrient content of commercial products was based on information provided by manufacturers. The body weight recorded daily was used to calculate daily energy and macronutrient intakes per kg of body weight. An Excel program to facilitate the calculations of the amounts of modular protein and fat supplements that were to be added to the fortified HM was developed and registered (Nona R, Cardoso M, Portuguese Directorate of Intellectual Property Services, IGAC-DSPI nr 480/2020, 26 February 2020). Energy and macronutrient intakes were compared to the ESPGHAN 2010 guidelines [19].

Anthropometry: Anthropometric parameters were assessed from birth to discharge. Body weight was measured by attending nurses at birth and daily during the hospital stay, with infants naked, using scales incorporated into the incubators or external automatic scales calibrated to the nearest gram [48]. The weight gain velocity (g/kg/day) was calculated using the Patel exponential model [49]. In the contemporary cohorts Subgroup 1b and Group 2 (Table 1), weekly crown–heel length and head circumference (HC) measurements were undertaken by the same observer (MC) to the nearest millimeter, and each considered measurement resulted from the average of three consecutive measurements. Crown–heel length was measured via the collaboration of two observers, using a rigid recumbent length board, and HC was measured encircling the supraorbital ridges and occipital protuberance using a non-stretchable measuring tape [48]. These measurements were used to calculate the length and HC gain velocities (cm/week). In the historical cohort (Subgroup 1a), the length and HC values recorded during the hospital stay were not considered, since methods for assuring the accuracy of these measurements were not included in the historical study design [33]. 

Body composition assessment: A single body composition assessment using displacement plethysmography (Pea Pod, Cosmed, Italy) was scheduled for up to one week after discharge for infants who maintained exclusive or predominant HM feeding. Fat mass (FM), FFM (%FM), and FFM percentage (%FFM) were provided automatically by the equipment. Using the aforementioned method, body length was measured by the same observer (MC), and this measurement was used to calculate the FM index (FMI) [50]. The main body composition outcome was adiposity, indicated by both the %FM and FMI. Excess adiposity was defined as %FM > 95th percentile for sex and PMA [51] and/or FMI > 90th percentile for PMA [50]. 

### 2.5. Sample Size Calculation

The study sample size was calculated to detect a difference of 2 g/kg/day in weight gain velocity, the main outcome, with a significance level of 0.05 and 80% power; a required sample of 68 infants was estimated, with 34 infants assigned to each group [34]. The assumption was based on the mean (SD) weight gain velocity of 10.1 (3.8) g/kg/d obtained in the historical cohort study [33] and 12.1 (1.6) g/kg/d reported by McLeod et al. [28] in a similar sample of preterm infants fed fortified HM based on HM measured macronutrient content. 

### 2.6. Statistical Analysis

The demographics and clinical characteristics of infants were described with frequencies (percentages) and with mean (SD: standard deviation) or median and interquartile range (IQR: 25th percentile–75th percentile), as appropriate. Nonparametric Chi-Square test, Fisher’s exact test, median test, and Mann–Whitney test were used, as needed.

Univariable and multivariable linear regression models were used to identify the variables which explained the body weight, length, and head circumference z-scores at discharge, and between birth and discharge (Δ z-scores), and the variability of body composition. Normality assumption of the residuals was verified using the Kolmogorov–Smirnov goodness-of-fit test with Lilliefors correction.

To study the association between independent variables and outcomes of excess adiposity (%FM > 95th percentile and FMI > 90th percentile), logistic regression models were used. For quantitative variables, logit linearity assumption (linear association with the logit of the outcome variable) was verified using generalized additive regression models.

Additionally, in order to investigate associations between independent variables and weight, length, and HC gain velocities, linear mixed effects regression models were applied. Interaction terms were considered in the multivariable study.

All the variables that attained a *p*-value ≤ 0.25 in the univariable analysis were candidates for the multivariable models.

A level of significance α = 0.05 was considered. Data analysis was performed using Stata (StataCorp, 2017, Stata Statistical Software: Release 15. College Station, TX: StataCorp LLC) and R (R: A Language and Environment for Statistical Computing, R Core Team, R Foundation for Statistical Computing, Vienna, Austria, year = 2022, http://www.R-project.org, accessed on 5 January 2023).

## 3. Results

### 3.1. Study Flowchart

As shown in the flowchart (Figure 1), 275 eligible infants were identified, but only 185 fulfilled the inclusion criteria: 115 in Group 1 (33 in Subgroup 1a plus 24 in Subgroup 1b) and 70 in Group 2. The reasons for not being recruited were parental refusal or inability to consent, hospital transfer, death, and formula feeding. No significant differences between recruited and not recruited infants were found regarding gestational age, birth weight z-score, twin prevalence, and prenatal steroids (Appendix A).

From the 185 infants recruited, 58 (50.4%) dropped-out in Group 1 and 12 (17.1%) in Group 2 (*p* < 0.001). The reasons for lost to follow-up were formula feeding in 64 infants and hospital transfer before completing the study exposure period in 6 infants. Therefore, 115 completed the study exposure period, 57 in Group 1 and 58 in Group 2. No significant differences were found between the infants completing the exposure period and those who were lost to follow-up regarding gestational age, birth weight z-score, twin prevalence, extreme prematurity, prenatal and postnatal steroids, severity risk, late-onset sepsis (LOS), NEC, intraventricular hemorrhage, bronchopulmonary dysplasia, and hospital stay. An exception was a higher severity risk in infants who were lost to follow-up (median SNAPPE II 15 vs. 10, *p* = 0.026) (Appendix A).

### 3.2. Characteristics of Infants Completing the Study

No significant differences were found between groups regarding demographic characteristics, birth weight, and morbidity, except for the infants of Group 2, who had a significantly higher prevalence of twins and LOS, and were significantly shorter and had lower head circumference at birth (Table 2).

The extremely preterm infants had a significantly higher severity risk, prevalence of LOS, and bronchopulmonary dysplasia than those who were born with more than 28 weeks’ gestation (Appendix A).

### 3.3. Nutritional Exposure

In both groups, the great majority of infants were fed MOM and a minority were fed DHM. The median (P_25_; P_75_) percentage of MOM volume administered during the hospital stay was 96.0% (71.2%; 100%) for Group 1 and 96.5% (80.8%; 99.2%) for Group 2. 

No significant differences existed between the two groups in the postnatal age and PMA at the beginning or end of exposure, in time intervals before exposure, and from the end of exposure to discharge, in the percentage of days of exposure in relation to the total days of enteral feeding, and the total days of hospital stay (Appendix A). In particular, no significant differences existed between Groups 1 and 2 in the median time of exposure to HM fortification (28.0 vs. 23.0 days, *p* = 0.072). 

A significant negative correlation was found between gestational age at birth and the number of days of exposure (r = −0.7, *p* < 0.001).

The proportion of exposure days in which the nutrient intakes did not reach the minimum or exceeded the maximum ESPGHAN 2010 recommendations [19] is presented in Appendix A. Regarding energy, PER, fat, and carbohydrate intakes, the proportion of days in which the maximum recommendations were exceeded was significantly higher in the fortification group based on measured HM macronutrient content, while the proportion of days in which the minimum recommended intakes of the same nutrients were not reached was significantly higher in the fortification group based on assumed HM macronutrient content.

In Table 3, the compliance of energy and macronutrient intakes during the exposure period with the ESPGHAN 2010 recommendations [19] is presented for each group and compared between groups. During the exposure period, infants fed fortified HM based on its measured macronutrient content received significantly higher energy, fat, and carbohydrate intake than those fed fortified HM based on its assumed macronutrient content. On the other hand, infants weighing ≥ 1 kg fed fortified HM based on its measured macronutrient content received significantly lower protein intake and those weighing < 1 kg received significantly lower PER intake than those fed fortified HM based on its assumed macronutrient content (Table 3). In a secondary analysis, it was found that the mean (SD) carbohydrate content of MOM administered was significantly higher, and administered for a longer exposure period, in the fortified HM based on its measured macronutrient content group than in the fortified HM based on its assumed macronutrient content group: mean (SD) 6.7 (0.9) g/dL administered for 1226 days vs. 6.5 (1.1) g/dL administered for 1412 days, *p* < 0.001. 

### 3.4. Anthropometry

#### 3.4.1. Body Weight z-Score

In both groups, the infants were born with negative body weight z-scores (Table 2), whose deviation increased up to discharge, without significant differences between groups (Table 4).

In the multivariable analysis, no significant association was found between the body weight Δ z-scores from birth to discharge and the method of HM fortification (*p* = 0.062). Whatever the fortification method, an increase in length z-score from birth to discharge was significantly associated with an increase in body weight z-score from birth to discharge (β estimate = 0.759; 95% CI 0.561, 0.957; *p* < 0.001).

#### 3.4.2. Weight Gain Velocity

By using the Patel method, the median weight gain velocity was significantly higher in infants fed fortified HM based on its measured macronutrient content than those fed fortified HM based on its assumed macronutrient content, both from birth to discharge (10.8 vs. 9.7 g/kg/d, *p* = 0.023) and from beginning of fortification to discharge (15.3 vs. 13.0 g/kg/d, *p* < 0.001). This comparison considered only the weight gain velocity estimates at the time of discharge.

Using a linear mixed effects model that considered daily estimates of weight gain velocity from birth to discharge, no association was found between them and the method of HM fortification (*p* = 0.158). This result is in line with the graph shown in Figure 2, which depicts the evolution of weight gain velocity in both groups with a lowess smoother superimposed. Whatever the fortification method, the weight gain velocity was significantly slower with a longer exposure (β estimate = −0.070; 95% CI −0.120, −0.019; *p* = 0.007) and faster in infants born at or above 28 weeks’ gestation (β estimate = 7.635; 95% CI 5.645, 9.624; *p* < 0.001). This means that for each additional day of exposure, a mean decrease of 0.070 g/kg/d in weight gain velocity occurred, and infants born at or above 28 weeks’ gestation gained weight at a mean velocity of 7.6 g/kg/d faster than extremely preterm infants.

#### 3.4.3. Length z-Score

In both groups, infants were born with negative length z-scores (Table 2), whose deviations increased up to discharge. The infants fed fortified HM based on its measured macronutrient content, despite being born significantly shorter, had a significantly smaller z-score decline and attained similar length z-score at discharge than those fed fortified HM based on its assumed macronutrient content (Table 4).

The multivariable analysis showed that a smaller decrease in mean length z-score variation from birth to discharge was associated with the measured-based HM fortification method (β estimate = 0.423; 95% CI 0.181, 0.664; *p* = 0.001) and with the decrease in the deviation of the body weight z-score from birth to discharge (β estimate = 0.560; 95% CI 0.414, 0.706; *p* < 0.001).

#### 3.4.4. Length Gain Velocity

In the univariable analysis, the length gain velocity had a significant positive association with the fortification method based on measured HM macronutrient content, gestational age, and birthweight, and a significant negative association with the number of days of exposure, LOS, and postnatal age. 

In the multivariable analysis, adjusted for gestational age, length gain velocity from birth to discharge had a significant positive association with the fortification method, with a faster mean increase of 0.20 cm/week (β-estimate = 0.20; 95% CI 0.126, 0.280; *p* < 0.001) using the fortification method based on measured HM macronutrient content.

#### 3.4.5. Head Circumference z-Score

In both groups, infants were born with negative HC z-scores (Table 2). The deviation of the HC z-score decreased at discharge in infants fed assumed-based fortified HM, remaining negative. In infants fed fortified HM based on its measured macronutrient content, despite being born with a significantly lower HC z-score, the HC z-score became positive at discharge, attaining a higher median HC z-score than those fed fortified HM based on its assumed macronutrient content (Table 4).

In the multivariable analysis, adjusting for the length z-score variation from birth to discharge (β estimate = 0.235; 95% CI 0.046, 0.425; *p* = 0.016), the fortification based on measured HM macronutrient content was significantly associated with a mean increase in the HC z-score from birth to discharge (β estimate = 0.955; 95% CI 0.672, 1.238; *p* < 0.001).

#### 3.4.6. Head Circumference Gain Velocity

In the univariable analysis, the HC gain velocity from birth to discharge had a significant positive association with the fortification method based on measured HM macronutrient content, gestational age, birthweight, length of stay, mean protein intake, and mean fat intake, and a significant negative association with the number of days of exposure and z-scores at birth of weight, length, and HC.

In the multivariable analysis, the HC gain velocity from birth to discharge was significantly positively associated with the fortification method, with a faster mean increase of 0.18 cm/week (β-estimate = 0.178; 95% CI 0.111, 0.246; *p* < 0.001) using the fortification method based on measured HM macronutrient content, adjusted for gestational age, HC z-score at birth, and length of stay.

### 3.5. Body Composition

Body composition was assessed in 74 (64.3%) infants after discharge, between 35 and 41 PMA: 38 in Group 1 and 36 in Group 2. The reasons that precluded body composition assessment were mainly due to the temporary closure of the Nutrition Laboratory, which occurred due to the COVID-19 pandemic, and parental refusal in the remaining cases.

No significant differences were found between infants with and without body composition assessment regarding gestational age, birth weight z-score, twin prevalence, extreme prematurity, prenatal and postnatal steroids, severity risk, intraventricular hemorrhage, bronchopulmonary dysplasia, NEC, and hospital stay. However, infants without body composition assessment had a higher occurrence of LOS (46.6% vs. 24.3%, *p* = 0.015), despite LOS not differing significantly between the HM fortification methods.

Infants fed fortified HM based on its measured macronutrient content, compared with those fed fortified HM based on its assumed macronutrient content, had significantly greater %FFM and lower FM and adiposity, indicated by both the %FM and FMI (Table 5).

Multivariable models were explored for body composition measurements, and specifically for adiposity excess.

The candidates for the body composition model (Table 5) were HM fortification method, number of exposure days, gestational age at birth, birth weight z-score, and sex, but no multivariable models were obtained.

The candidates for the adiposity excess model (%FM > 95th percentile and FMI > 90th percentile) were HM fortification method, number of days of exposure, gestational age at birth, and mean energy, protein, carbohydrate, and fat intakes.

In the univariable logistic regression for the outcome of excess adiposity (%FM > 95th percentile), considering the fortification based on assumed HM macronutrient content as a reference category, weak evidence was found that infants fed fortified HM based on its measured macronutrient content had 74% lower odds of excess adiposity (OR estimate = 0.26; 95% CI 0.06, 1.02; *p* = 0.053). No multivariable models were found for excess adiposity.

## 4. Discussion

In this cohort study, it was found that infants born at less than 33 weeks’ gestation fed fortified HM based on its measured macronutrient content received significantly higher mean energy, fat, and carbohydrate intakes than those fed fortified HM based on its assumed macronutrient content. On the other hand, infants weighing ≥ 1 kg fed fortified HM based on its measured macronutrient content received significantly lower mean protein intake and those weighing < 1 kg received significantly lower mean PER than those fed fortified HM based on its assumed macronutrient content. Infants fed fortified HM based on its measured macronutrient content had significantly better weight gain, length, and head growth at discharge. Body composition assessment at or close to TEA showed that infants who had been fed fortified HM based on its measured macronutrient content had significantly lower adiposity and greater lean body mass.

### 4.1. Nutrient Intake According to the Fortification Method

In this observational study, the HM fortification prescription was left to clinical discretion. As such, a higher variability in nutrient intake would be expected than when following a fixed interventional study protocol.

The variability in energy and macronutrient intakes in the fortification based on assumed HM macronutrient content is in line with previous reports [18,28,52], and it was mainly due to intakes not achieving the minimum recommended amounts [19]. Overestimation of HM energy and macronutrient content was reported when fortification relies on assumed HM macronutrient content [18,32,53]. 

Individualized HM fortification was reported to provide the recommended nutrient intakes, with less macronutrient intake variability [52,54]. However, using fortification based on measured HM macronutrient content, we found an important variation in energy and macronutrient intakes, predominantly exceeding the maximum recommended amounts [19]. Rochow et al. [18] also reported excessive macronutrient intakes when using target fortification with added modular supplements, attempting to comply with ESPGHAN 2010 guidelines. Similarly, we found that infants fed fortified HM based on its measured macronutrient content received recommended mean intakes of energy, protein, and carbohydrates, but a mean fat intake (6.7 mg/kg/d) higher than the maximum recommended (6.6 mg/kg/d). In particular, infants weighing < 1 kg fed fortified HM based on its measured macronutrient content have received a median PER intake lower than the minimum recommended (3.3 g/100 kcal) [19] due to disproportionately high energy intake in relation to the recommended protein intake. Nevertheless, the mean fat intake and the median PER intake were found to be within the range of 4.8–8.1 g/kg/d and 2.8–3.6 g/100 kcal, respectively, of the updated ESPGHAN 2022 guidelines for enteral nutrition in preterm infants, available after the study was concluded [55].

Since both fortification methods used the same fortifier without the addition of a modular carbohydrate supplement, a secondary analysis was performed to explain the differences in carbohydrate intake, and it was found that in the fortified HM based on its measured macronutrient content group, the carbohydrates were significantly denser in the MOM administered during the exposure period.

### 4.2. Body Weight Gain

In this study, to assess inadequate nutrition in preterm infants, we chose the body weight z-score decline and the weight gain velocity, since they are considered the most reliable metrics [56].

The infants of both groups were born with negative body weight z-scores, with a deviation that increased up to discharge, without significant differences between the groups. The multivariable analysis showed that the body weight Δ z-score between birth and discharge was not associated with the method of HM fortification; rather, it was associated with an increase in length z-score in the same period. In fact, body length reflects more specifically the skeletal growth and fat-free mass, while body weight is composed of the weight of multiple organs and tissues [56,57].

Z-score differences calculated from birth weight-based charts were criticized for not accurately reflecting the growth rate of very preterm infants [58,59]. Otherwise, the weight gain velocity over the previous days is reported to be more sensitive with respect to identifying changes in growth, with a minimum of a 5–7-day period required for a precise calculation [60]. It was suggested that 15–20 g/kg/d is the optimal weight velocity in infants born at 23–36 weeks’ gestation, approximating to the mean estimates of fetal growth [61,62].

Compared with the conventional two-point assessment method, an exponential model was proposed by Patel et al. [49] as a more accurate estimate of weight gain velocity, as it is not affected by decreasing birth weight and increasing length. Using the Patel model, we found that the median weight gain velocity at discharge was significantly faster in the fortification group based on measured HM macronutrient content than in the fortification group based on assumed HM macronutrient content, either from birth (10.8 vs. 9.7 g/kg/d) or from the beginning of fortification (15.3 vs. 13.0 g/kg/d).

However, using linear mixed effects models that considered the correlation structure between the longitudinal estimates, no difference in weight gain velocity was found between the groups. In line with this, McLeod et al., using the Patel exponential model, did not find differences in weight gain velocity between the assumed- and measured-based HM fortification methods [28].

The multivariable analysis with stratification by gestational age showed that, whatever the fortification method, weight gain velocity was significantly faster in infants who were born at ≥28 weeks’ gestation than in extremely preterm infants, an expected finding. The slower weight gain velocity in extremely premature infants may be related to significantly higher severity risk and LOS and higher incidence of bronchopulmonary dysplasia, morbidities that are inherent to extreme prematurity [63]. An association was also found between a longer exposure to fortification and a significantly slower weight gain velocity. The deceleration in weight gain velocity with the duration of fortification may reflect a progressive overcoming of cumulative nutrient deficits in malnourished infants, whose nutritional status improved as better nutritional support became possible on a regular basis [64].

### 4.3. Linear Growth

Infants of both groups were born with negative length z-scores and had linear growth faltering up to discharge, indicated by a decline in length z-scores. Despite the fact that the infants fed fortified HM based on its measured macronutrient content were born significantly shorter than those fed fortified HM based on its assumed macronutrient content, they had a significantly smaller length Δ z-score decline, with both groups attaining similar mean length z-scores at discharge. This is consistent with a significant positive association between the fortification method based on measured HM macronutrient content and the length gain velocity, adjusted for gestational age, being 0.2 cm/week faster than using fortification based on assumed HM macronutrient content. In a systematic review comparing the effect of individualized fortification (targeted or adjustable) with standard non-individualized HM fortification on growth, infants fed individualized fortified HM had a length gain velocity 0.3 cm/week faster during the intervention [15]. This is clinically relevant, since superior linear growth was reported to be associated with better brain development [56].

### 4.4. Head Growth

The HC measurements reflect brain size and head growth, which is an important indicator of neurodevelopment outcomes in preterm infants [56,65].

Infants in both groups were born with negative HC z-scores. In the fortification group based on assumed HM macronutrient content, HC z-scores declined up to discharge. Despite the fact that the infants fed fortified HM based on its measured macronutrient content were born with significantly lower mean HC z-scores, their HC z-scores increased up to discharge, attaining a higher mean z-score than in the infants fed fortified HM based on its assumed macronutrient content. Consistently, the multivariable analysis revealed that the increase in HC z-scores from birth to discharge was significantly associated with the fortification based on measured HM macronutrient content, adjusted for length z-score variation in the same period.

The HC gain velocity was significantly associated with fortification based on measured HM macronutrient content, adjusted for gestational age, HC z-score at birth, and length of stay, and was 0.18 cm/week faster than using fortification based on assumed HM macronutrient content. In the aforementioned systematic review, preterm infants fed individualized fortified HM had, during the intervention, an HC velocity 0.10 cm/week faster than the standard non-individualized fortified HM [15].

### 4.5. Body Composition

In a systematic review with meta-analysis, it was concluded that preterm infants at TEA have greater %FM and less FFM than those of infants born at term [20]. This increase in adiposity occurs even when nutrition is prescribed, according to the ESPGHAN 2010 guidelines [21]. The mentioned differences in body composition at TEA were reported to dissipate by 3–5 months’ corrected age [22].

The body composition assessed after discharge at late preterm or term PMA (35 to 41 weeks) showed, via univariable analysis, that infants who had been fed fortified HM based on its measured macronutrient content had significantly lower fatness, as indicated by FM and adiposity (%FM and FMI), and greater leanness, as indicated by %FFM, compared with those fed fortified HM based on its assumed macronutrient content. In the univariable logistic regression, weak evidence was found that the infants fed fortified HM based on its measured macronutrient content had lower odds of excess adiposity; no multivariable models were obtained, rendering it impossible to control for other variables reported to affect fatness. This greater %FFM in infants fed fortified HM based on its measured macronutrient content is consistent with their faster linear growth, another indicator of lean mass [66]. This is clinically relevant, since in preterm infants at TEA, it is reported that a greater leanness is associated with brain size [66,67].

These results for body composition raise two questions. First, why did infants who were fed fortified HM based on its assumed macronutrient content and receiving lower than recommended energy and fat intakes had greater adiposity at near TEA compared with those fed fortified HM based on its measured macronutrient content who received higher energy and fat intakes? Second, why did infants who were fed fortified HM based on its measured macronutrient content had lower adiposity and greater leanness at near TEA after receiving in-hospital energy and protein intakes according to ESPGHAN 2010 guidelines [19], but mean fat higher than the maximum intake recommended, as well as a median PER (in infants weighing < 1 kg) lower than the minimum intake recommended?

We speculate that a mechanism could justify these findings. It has been described that in very preterm infants, postnatal age and PMA are strong predictors of FM and %FM at 34–37 weeks PMA [68]. As the rapid accumulation of fetal fat begins by 32–33 weeks’ gestation, after a preterm birth interrupting the constant nutrient supply of the placenta, the ability for fat accretion may be an adaptive postnatal mechanism protecting against the risk of low nutrient availability [68,69]. Using the fortification regimen based on measured HM macronutrient content, higher in-hospital energy, protein, and carbohydrate intakes were provided, which could have reduced excessive fat accumulation at this age [68].

### 4.6. Limitations

Certain limitations of this study should be acknowledged.

This is an observational study that assessed the effectiveness of two methods of HM fortification in real-life clinical practice, where nutritional prescription was at the clinical discretion of physicians. The findings of this study do not substitute those of a randomized clinical trial assessing the efficacy of evaluated nutritional strategies.

Infants lost to follow-up had significantly higher severity risk than those completing the study period. On the other hand, infants fed fortified HM based on its measured macronutrient content were born significantly shorter and with lower HC and had significantly higher severity risk, prevalence of twins, and LOS than infants fed fortified HM based on its assumed macronutrient content. Although these latter differences may be biases, they did not preclude the group fed fortified HM based on its measured macronutrient content from having better growth outcomes despite having more unfavorable factors.

To accurately accomplish what the study set out to do, with recommendations for enteral nutrition in preterm infants, the ideal would have been to guide fortification based on daily analysis of MOM composition, but such frequency of analysis would be impractical as it would be overly laborious and time consuming [18,28]. Moreover, an excess of MOM volume required for frequent analyses would reduce its availability to feed the infants most in need. The reported weekly analysis of MOM has been the better possible alternative to achieve the study’s purpose [53].

Body composition was based on a single assessment soon after discharge, rather than providing insight on previous changes and their relationship to in-hospital nutritional support. Nevertheless, our data can be compared with those from most other similar studies, which have focused on the body composition of preterm infants at TEA [20,70].

### 4.7. Strengths

In the contemporary cohort, including the entire sample of fortification based on measured HM macronutrient content and part of the sample of fortification based on assumed HM macronutrient content, nutrient intakes were accurately recorded daily by the same observer and the amounts of modular protein and fat supplements added to the fortified HM were accurately calculated using a specifically developed informatics tool.

Growth assessment during the hospital stay relied on weight, length, and HC Δ z-scores and their velocities, which are considered the better metrics to evaluate growth in preterm infants.

To estimate weight gain velocity, in addition to using an accurate exponential method, which is superior to the conventional two-point assessment method, refined linear mixed effects regression models that consider the correlation structure between longitudinal measurements were used to analyze those estimates, providing differentiated results.

A better body composition profile was found associated with mean fat higher than the maximum intake recommended and median PER (in infants weighing < 1 kg) lower than the minimum intake recommended by the contemporary ESPGHAN 2010 guidelines [19]. Importantly, these nutrient intakes turned out to be within the range of the latter updated ESPGHAN 2022 guidelines [55], supporting the advantages of these new enteral nutrition recommendations for preterm infants.

## 5. Conclusions

In this cohort study of infants born at less than 33 weeks of gestation, those fed fortified HM based on its measured macronutrient content had significantly better weight gain, length, and head growth at discharge, and lower adiposity and greater lean mass at or near TEA. This body composition profile was associated with energy and protein intakes in the range of the ESPGHAN 2010 guidelines but mean fat higher than the maximum intake recommended and median PER (in infants weighing < 1 kg) lower than the minimum intake recommended. These nutrient intakes were later revealed to be within the range of the updated ESPGHAN 2022 guidelines.

Further high-quality trials are needed to confirm if the enteral nutritional regimen used in this observational study is more appropriate for better growth and body composition in preterm infants.

## Figures and Tables

**Figure 1 nutrients-15-01533-f001:**
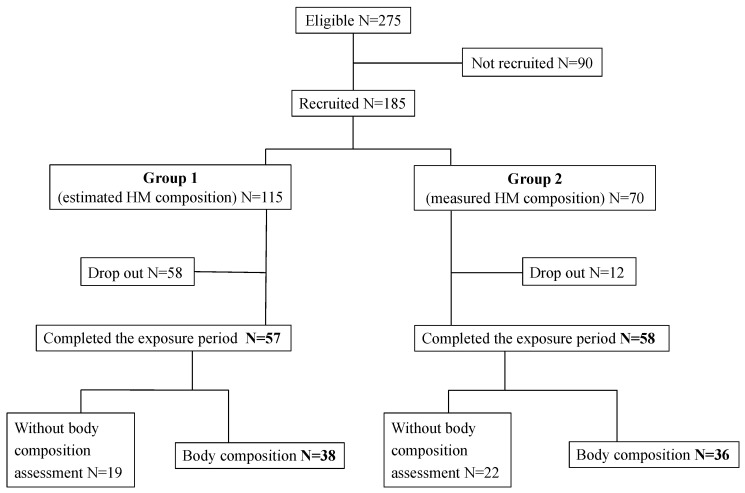
Study flowchart.

**Figure 2 nutrients-15-01533-f002:**
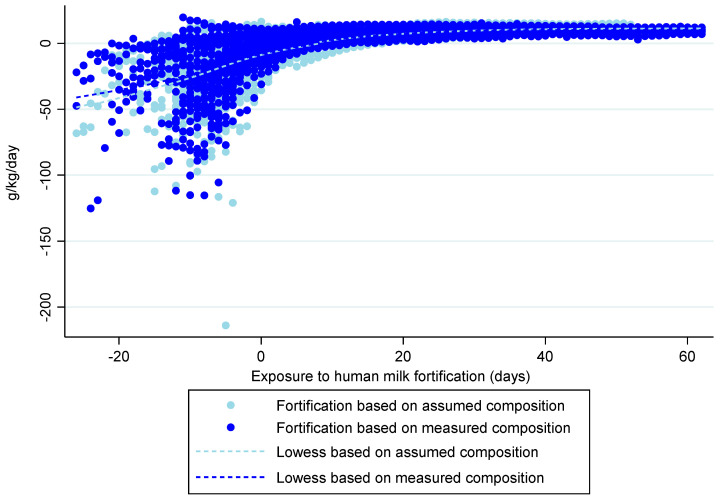
Weight gain velocity model in both groups, from birth to discharge, using a lowess smoother. Day ‘0’ indicates the starting day of exposure to human milk fortification.

**Table 1 nutrients-15-01533-t001:** Periods of recruitment of the infants fed either fortified HM based on its assumed macronutrient content (Group 1) or fortified HM based on its measured macronutrient content (Group 2).

Group 1	Group 2
- Subgroup 1a: Historical cohort—recruitment from February 2014 to February 2015	- Recruitment from 14 July 2020 to 31 May 2021
- Subgroup 1b: Contemporary cohort—recruitment from 1 February to 13 July 2020	

HM—human milk.

**Table 2 nutrients-15-01533-t002:** The characteristics of infants that completed the exposure period (*n* = 115). Group 1—fortified HM based on its assumed macronutrient content. Group 2—fortified HM based on its measured macronutrient content.

	Group 1	Group 2	*p*-Value
	*n* = 57	*n* = 58	
Gestational age in weeks, mean (SD)	29.6 (1.99)	29.7 (2.31)	0.850
Females, *n* (%)	27 (47.4)	23 (39.7)	0.412
Twins, *n* (%)	15 (26.3)	24 (41.4)	0.047
Extremely preterm, *n* (%)	14 (24.6)	12 (20.7)	0.314
Prenatal steroids, *n* (%)	55 (96.5)	53 (91.4)	0.286
Birth weight z-score, mean (SD)	−0.11 (0.70)	−0.19 (0.66)	0.691
Birth length z-score, mean (SD)	−0.48 (0.512)	−1.07 (0.740)	0.001
Birth HC z-score, mean (SD)	−0.83 (0.90)	−1.60 (0.778)	<0.001
Postnatal steroids, *n* (%)	1 (1.8)	4 (6.9)	0.102
Severity risk (SNAPPE II), median (P_25_; P_75_)	10 (0.22)	10 (0.27)	0.785
Late-onset sepsis, *n* (%)	12 (21.1)	25 (43.1)	0.001
Necrotizing enterocolitis III, *n* (%)	0	0	-
Intraventricular hemorrhage I, *n* (%)	2 (3.5)	0	0.128
Bronchopulmonary dysplasia, *n* (%)	4 (7.0)	6 (10.3)	0.264

HC—head circumference, HM—human milk, SD—standard deviation. Student-*t* test, Chi-Square test, Fisher’s exact test, median test, or Mann–Whitney test, as appropriate.

**Table 3 nutrients-15-01533-t003:** Energy and macronutrient intake in infants completing the study exposure period (*n* = 115). Group 1—fortified HM based on its assumed macronutrient content. Group 2—fortified HM based on its measured macronutrient content.

	Recommended [19]	Group 1	Group 2	*p*-Value
		*n* = 57	*n* = 58	
Total energy intake, kcal/kg/d; mean (SD)	110–135 kcal/kg/d	121.3 (25.3)	127.9 (17.4)	<0.001
Protein intake, g/kg/d; median (P_25_; P_75_)	All	4.2 (3.7; 4.7)	4.1 (3.7; 4.6)	0.109
	<1 kg body weight: 4.0–4.5 g/kg/d	3.9 (3.2;4.8)	4.1 (3.3; 4.9)	0.162
	1–1.8 kg body weight: 3.5–4.0 g/kg/d	4.2 (3.8; 4.6)	4.1 (3.8; 4.5)	0.004
PER intake, g/100 kcal; median (P_25_; P_75_)	All	3.5 (3.2; 3.9)	3.4 (3.0; 3.8)	0.001
	<1 kg body weight: 3.6–4.1	3.5 (3.2; 4.0)	3.3 (3.0; 3.7)	<0.001
	1–1.8 kg body weight: 3.2–3.6	3.4 (3.2; 3.8)	3.5 (3.1; 3.9)	0.315
Fat intake, g/kg/d; mean (SD)	4.8–6.6 g/kg/d	5.5 (1.54)	6.7 (1.67)	<0.001
Carbohydrate intake, g/kg/d; mean (SD)	11.6–13.2 g/kg/d	11.9 (1.93)	13.0 (1.90)	<0.001

HM—human milk; PER—protein-to-energy ratio. Student-*t* test, median test, or Mann–Whitney test, as appropriate.

**Table 4 nutrients-15-01533-t004:** Body weight, length, and head circumference Δ z-scores from birth to discharge (*n* = 115). Group 1—fortified HM based on its assumed macronutrient content. Group 2—fortified HM based on its measured macronutrient content.

	Group 1	Group 2	*p*-Value
	*n* = 57	*n* = 58	
Body weight z-score at discharge, mean (SD)	−1.73 (0.92)	−1.73 (0.870)	0.693
Δ body weight z-score, median (P_25_; P_75_)	−1.64 (−1.99; −1.28)	−1.47 (−1.79; −1.10)	0.565
Length z-score at discharge, median (perc. 25; perc. 75)	−1.47 (−1.685; −0.970) †	−1.49 (−2.135; −0.830)	0.744
Δ length z-score, median (P_25_; P_75_)	−0.71 (−1.69; −0.37) †	−0.27 (−2.14; −0.04)	0.005
HC z-score at discharge, median (P_25_; P_75_)	−1.17 (−1.70; −0.63) †	−0.80 (−1.390; −0.02)	0.102
Δ HC z-score, median (P_25_; P_75_)	−0.31 (−0.59; 0.04) †	0.75 (0.41; 1.17)	<0.001

† Only assessed in 24 infants of the contemporary cohort fed assumed-based fortified human milk. *p*-values obtained via linear regression models; HC—head circumference, HM—human milk, SD—standard deviation. Student-*t* test, median test, or Mann–Whitney test, as appropriate.

**Table 5 nutrients-15-01533-t005:** Single assessment of body composition after discharge, between 35–41 weeks postmenstrual age (*n* = 74). Group 1—fortified HM based on its assumed macronutrient content. Group 2—fortified HM based on its measured macronutrient content.

	Group 1	Group 2	*p*-Value
	*n* = 38	*n* = 36	
FM (g), median (P_25_; P_75_)	360.1 (260.0; 483.3)	269.5 (169.1; 413.9)	0.026
FFM (g), median (P_25_; P_75_)	2155.8 (1854.8; 2529.8)	2034.9 (1834.5; 2253.4)	0.173
%FM, median (P_25_; P_75_)	14.5 (12.2; 17.3)	12.6 (8.3; 14.9)	0.021
FMI, median (P_25_; P_75_)	1.8 (1.4; 2.3)	1.4 (0.9; 1.8)	0.004
%FFM, median (P_25_; P_75_)	85.5 (82.7; 87.6)	87.4 (85.1; 91.0)	0.012

Mann–Whitney test; FM—fat mass; %FM—percentage FM; FFM—fat-free mass; %FFM—percentage FFM; FMI—fat mass index; HM—human milk. Student-*t* test, median test, or Mann–Whitney test, as appropriate.

## Data Availability

The data presented in this study may be available on request from the corresponding author. According to the institutional policy, the data are not publicly available, complying with the confidentiality and protection of personal data.

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
