# Peer review of "Individualized Fortification Based on Measured Macronutrient Content of Human Milk Improves Growth and Body Composition in Infants Born Less than 33 Weeks: A Mixed-Cohort Study"

_nutrients, 2023, doi:10.3390/nu15061533_

Round 1
Reviewer 1 Report
This manuscript reports human milk fortification practices for very preterm infants. Overall, the topic is worthwhile to be explored and can provide much needed insight to better understand the nutritional needs of the preterm infant. However, there are several concerns that need to be addressed before this manuscript may be publishable.
Further, line Numbers are not provided in the manuscript to enable the reviewer to exactly pinpoint the issue of the concern to the author. Language revisions are needed, preferably conducted by a native English speaker. There are various issues with use of past and present tense, and the convoluted manner of writing which need to improve so the reader is enabled to follow the author’s intention. Please ensure all abbreviations are explained according to the journal’s guidelines.
Title:
The title does not translate the purpose of this paper in a clear manner.
1. What was measured? Based on the title the fortification was measured. But what components? What is “the quality of growth”? The preterm infants simply showed better growth I assume? Please chose a title that clearly states what was done.
2. Given that the fortification of importance for this manuscript was only done for protein and fat (as per description), this should be included in the title to clearly describe to reader what this manuscript is all about. The “measured composition” is very general, and you are not measuring composition with the Miris, but macronutrients. Since macronutrients are the only nutrients of concern here, this should be clarified, otherwise it is misleading.
Abstract:
1. First sentence: again very general. HM fortification for which compounds. HM contains a myriad of compounds, yet you have not specified your HM compounds of interest and you merely look at 3. Please edit accordingly.
2. You are not measuring human milk composition; you are measuring macronutrient content. Please edit accordingly. See comment above.
3. I understand that the abstract is quite limited in word count. However, there should be some explanation to your main outcome. Just using the terms “assumed-fed” and “measured-fed” does not mean much to the reader at this point.
4. Thew HM analyzer should be named, it does not really increase the word count much. Again, you are not measuring human milk composition.
5. The length of the treatment should be included in the abstract as again, this is a main factor for your primary outcome. In fact it is not clearly defined anywhere in the manscuript.
Introduction
1. Please define “very-preterm infants” in the first paragraph. I assume < 33 wks estation?
2. 3rs paragraph: What nutrients are found in those multi-nutrient fortifiers? Are they more focused on macronutrients/energy or also micronutrients? Please explain.
3. 3rd paragraph: Adding CHO, protein, and fats are added to achieve desirable macronutrient targets as you are only referring to macronutrients.
4. 5th paragraph: adequate nutrition for macronutrient and energy intake I assume? You are not achieving adequate overall nutrition when fortifying with macronutrients when the infant is also deficient in micronutrients. Please clarify.
Methods:
1. Section 2.2:
a. why is it important that the mothers have 2 consecutive pregnancy that ended with preterm delivery?
b. Please define the exposure period and explain in more detail.
c. What is the connection between the shortage of supplement and stopping the new nutritional protocol. Based on your explanation, my understanding is that the difference between the groups is simply whether the milk was measured with Miris or not. How is the ability to measure the macronutrients related to the availability of a supplemented? Please clarify and explain in more detail.
d. How was the sample size estimated? What is achieved with 68 infants? Please explain.
2. Table 2: I assume subgroup 1a is not still recruiting until 2025?
Results:
1. P7: you state that there weren’t any significant differences between the groups expect for.... Thus, you have significant differences in the characteristics, and it is not just for infants in group 2 as there can only be a difference when comparing between the groups. Language revisions most likely will take care of this, as recommended above.
2. Section 3.3:
a. Please double check your data. How come that your median is equal to your P75 in both cases?
b. You are stating that infants weighing < 1kg fed measured-based fortified HM received less protein intake. OK, but how did you measure this as you have not described any intake measurements. Without intake data such as D2M or test weighing you actually don’t know how much the infant consumed. If such data was used, please describe the protocol for such data in the respective section. Without such data, you only have the info on how much the milk was fortified, and your conclusions cannot be derived due to intake data.
3. Table 3: continues the same issue with intake. You need to include intake data if you want to draw conclusions based on intake and not on fortification level. The latter is the one described in the manuscript; intake is not. Intake and fortification level are not interchangeable.
4. P9: If I understand correctly, you describe that infants born <28 wks had a lower growth velocity than infant born >28wks? Did you take into account the difference of developmental stages of these infants born at different times of gestation? How would that affect the growth velocity? If the infant is more mature at birth, that may already be a beneficial factor for its growth. Something potentially worth adding in the Discussion.
Discussion:
1. 1st paragraph: again, please describe your intake data.
2. Section 4.1.
a. P11: which nutrients were reported as excessive in ref 27?
b. P12: how much denser was MOM in CHO?
Author Response
This manuscript reports human milk fortification practices for very preterm infants. Overall, the topic is worthwhile to be explored and can provide much needed insight to better understand the nutritional needs of the preterm infant. However, there are several concerns that need to be addressed before this manuscript may be publishable.
Further, line Numbers are not provided in the manuscript to enable the reviewer to exactly pinpoint the issue of the concern to the author. Language revisions are needed, preferably conducted by a native English speaker. There are various issues with use of past and present tense, and the convoluted manner of writing which need to improve so the reader is enabled to follow the author’s intention. Please ensure all abbreviations are explained according to the journal’s guidelines.
Response: Thank you for the insightful questions and suggestions.
In the submitted manuscript, which is available on the Nutrients platform, lines are numbered throughout the entire manuscript.
Using MDPI services, the authors have ordered English editing service and a certificate (#58610) was issued. Nevertheless, while revising the manuscript an effort was made by the authors to only use the past tense and parts of the manuscript were rewritten to be clearer.
All abbreviations are now explained in the revised manuscript.
Title:
The title does not translate the purpose of this paper in a clear manner.
- What was measured? Based on the title the fortification was measured. But what components? What is “the quality of growth”? The preterm infants simply showed better growth I assume? Please chose a title that clearly states what was done.
- Given that the fortification of importance for this manuscript was only done for protein and fat (as per description), this should be included in the title to clearly describe to reader what this manuscript is all about. The “measured composition” is very general, and you are not measuring composition with the Miris, but macronutrients. Since macronutrients are the only nutrients of concern here, this should be clarified, otherwise it is misleading.
Response: Thank you for the suggestion to provide a clearer title. Accordingly, the title was reformulated to “Individualized Fortification Based on Measured Macronutrient Content of Human Milk Improves Growth and Body Composition of Infants Born Less Than 33 Weeks: A Mixed-Cohort Study”.
Abstract:
- First sentence: again very general. HM fortification for which compounds. HM contains a myriad of compounds, yet you have not specified your HM compounds of interest and you merely look at 3. Please edit accordingly.
- You are not measuring human milk composition; you are measuring macronutrient content. Please edit accordingly. See comment above.
- I understand that the abstract is quite limited in word count. However, there should be some explanation to your main outcome. Just using the terms “assumed-fed” and “measured-fed” does not mean much to the reader at this point.
- The HM analyzer should be named, it does not really increase the word count much. Again, you are not measuring human milk composition.
- The length of the treatment should be included in the abstract as again, this is a main factor for your primary outcome. In fact it is not clearly defined anywhere in the manuscript.
Response: Thank you for the comments and suggestions.
The Abstract was modified accordingly, replacing the terms “assumed feed” and “measured feed” with more precise terms, although requiring more words. Not to exceed too much the Abstract word count, in the objective it is specified that 'fortification was based on assumed or measured HM macronutrient content’, but further on it is only stated that 'Fortification was based on assumed or measured HM content’, it being understood that the measurement concerned macronutrients.
In the revised Abstract, the HM analyzer model and the fortification exposure period were included as suggested.
To comply with the recommended suggestions without eliminating essential information already mentioned, the word count has been slightly increased.
Introduction:
- Please define “very-preterm infants” in the first paragraph. I assume < 33 wks gestation?
Response: The second Reviewer suggested reducing the Introduction section eliminating non-essential information, and the sentence where “very-preterm infants” was included was eliminated. Nevertheless, very-preterm infant is defined as an infant born with less than 32 weeks of gestation.
- 3rd paragraph: What nutrients are found in those multi-nutrient fortifiers? Are they more focused on macronutrients/energy or also micronutrients? Please explain.
Response: Thank you for the question. In the revised manuscript it was added “…HM multi-nutrient fortifiers are used, containing protein, carbohydrates, minerals, and vitamins, some fortifiers also containing fat [2,10,11]”
- 3rdparagraph: Adding CHO, protein, and fats are added to achieve desirable macronutrient targets as you are only referring to macronutrients.
Response: Thank you for the suggestion. Accordingly, “nutrients” was replaced with “macronutrients”.
- 5thparagraph: adequate nutrition for macronutrient and energy intake I assume? You are not achieving adequate overall nutrition when fortifying with macronutrients when the infant is also deficient in micronutrients. Please clarify.
Response: Thank you for the suggestion. Accordingly, “adequate nutrition” was replaced with “adequate intake of energy and macronutrients”.
Methods:
- Section 2.2:
- why is it important that the mothers have 2 consecutive pregnancy that ended with preterm delivery?
Response: Thank you very much for the question and the opportunity to clarify. The term “consecutive” refers to sampling of consecutive infants who met the inclusion criteria, until the estimated sample size was reached. In the revised manuscript it is clearer stated “The eligibility criteria [34] were those used in the historical cohort study [33], with sampling of consecutive infants born with less than 33 weeks of gestation…”
- Please define the exposure period and explain in more detail.
Response: Thank you for the question and the opportunity to clarify. The statement was reformulated to be clearer “The exposure period was defined by feeding fortified HM for at least 2 weeks. Recruited infants were dropped-out if they were fed formula for two or more days > 12.5% daily volume intake or were transferred or deceased before completing the exposure period”.
- What is the connection between the shortage of supplement and stopping the new nutritional protocol. Based on your explanation, my understanding is that the difference between the groups is simply whether the milk was measured with Miris or not. How is the ability to measure the macronutrients related to the availability of a supplemented? Please clarify and explain in more detail.
Response: Thank you for the question and the opportunity to clarify. The new institutional nutritional protocol was not limited to the measurement of HM macronutrient content. As stated in Section 2.2, the new nutritional protocol also included the individualization of fortification by adding HM fortifier and eventually modular protein and fat supplements, guided by the measurement of HM macronutrient. As shortage of one nutritional supplement (modular protein) occurred, starting the new protocol without the modular protein available would not comply with the institutional protocol. This is better explained in the revised Section 2.2.
For unforeseen reasons, the modular protein shortage lasted longer than we expected.
- How was the sample size estimated? What is achieved with 68 infants? Please explain.
Response: Thank you for the question. The sample size calculation is explained in the published study protocol which is cited in the manuscript. Nevertheless, to be more readily available, the sample size calculation was included in the new Section 2.5 of the revised manuscript.
- Table 2: I assume subgroup 1a is not still recruiting until 2025?
Response: Thank you for pointing out the error that was corrected: it is ‘2015’ instead of ‘2025’, according to the text description.
Results:
- P7: you state that there weren’t any significant differences between the groups expect for.... Thus, you have significant differences in the characteristics, and it is not just for infants in group 2 as there can only be a difference when comparing between the groups. Language revisions most likely will take care of this, as recommended above.
Response: Thank you very much for the question and the opportunity to clarify. In fact, the first paragraph of Section 3.2 was confusing, and it was completely rephrased to be clearer.
- Section 3.3:
- Please double check your data. How come that your median is equal to your P75 in both cases?
Response: Thank you very much for the question and the opportunity to clarify and correct an error. In the revised manuscript, the paragraph was completely rephrased, and the error corrected.
- You are stating that infants weighing < 1kg fed measured-based fortified HM received less protein intake. OK, but how did you measure this as you have not described any intake measurements. Without intake data such as D2M or test weighing you actually don’t know how much the infant consumed. If such data was used, please describe the protocol for such data in the respective section. Without such data, you only have the info on how much the milk was fortified, and your conclusions cannot be derived due to intake data.
Response: Thank you for the question. Macronutrient intakes were calculated as follows: For each infant, daily HM (MOM or DHM) volume intakes were recorded. To calculate macronutrient (g/mL) and energy (kcal/mL) densities of fortified HM, the macronutrient content of HM administered was recorded daily, as well as the macronutrient content of the added HM fortifier, modular protein, and MCT oil. The measured HM macronutrient content was used for calculations in the ‘measured HM macronutrient content group’ and the estimated HM macronutrient content based on literature data was used in the ‘assumed HM macronutrient content group’. The macronutrient content of commercial products was based on information provided by manufacturers. The body weight recorded daily was used to calculate daily energy and macronutrient intakes per kg of body weight. Macronutrient intakes were calculated stratifying according to the infants’ body weight (< or ≥ 1 Kg body weight) and gestational age (< or ≥ 28 weeks) categories. This information was added in detail in the revised 2.4 section.
- Table 3: continues the same issue with intake. You need to include intake data if you want to draw conclusions based on intake and not on fortification level. The latter is the one described in the manuscript; intake is not. Intake and fortification level are not interchangeable.
Response: Please see the response to the previous question.
- P9: If I understand correctly, you describe that infants born <28 wks had a lower growth velocity than infant born >28wks? Did you take into account the difference of developmental stages of these infants born at different times of gestation? How would that affect the growth velocity? If the infant is more mature at birth, that may already be a beneficial factor for its growth. Something potentially worth adding in the Discussion.
Response: Thank you for the question. The study was not designed to assess the effect of gestational age with 1-week precision. To refine the analysis, we have just stratified the gestational age, categorizing by the usual 28-week threshold. We acknowledged that more mature infants at birth may already have a beneficial factor for their growth. In Discussion (section 4.2), we interpreted the slower weight gain velocity expected in extremely preterm infants (< 28-weeks) as related with higher severity risk inherent to extreme prematurity and morbidities such as late-onset sepsis and higher incidence of bronchopulmonary dysplasia observed in our infants.
Discussion:
- 1stparagraph: again, please describe your intake data.
Response: A reformulated and detailed description of intake data calculation is now stated in Methods, section 2.4.
- Section 4.1.
- P11: which nutrients were reported as excessive in ref 27?
Response: Thank you the question and the opportunity to clarify. The ‘nutrients’ refers to ‘macronutrients’, as corrected in the revised manuscript.
- P12: how much denser was MOM in CHO?
Response: Thank you for the question and the opportunity to clarify. To evaluate the effect of providing a higher CHO content in the fortified HM based on its measured macronutrient content group, we considered not only the CHO density of MOM itself, but also the duration of MOM administration. As stated in section 3.3, fortified MOM based on its measured macronutrient content provided a significantly higher CHO density over a longer exposure period than in the fortified MOM based on its assumed macronutrient content group: mean (SD) 6.7 (0.9) g/dL administered for 1226 days vs. 6.5 (1.1) g/dL administered for 1412 days, p <0.001. This is better clearer explained in the revised manuscript.

Reviewer 2 Report
Authors performed an observational study to assess whether the implemented fortification method based on measured HM composition improves the energy and macronutrient intake and the quality of growth in infants born less than 33 weeks’ gestation, compared with HM fortification based on assumed HM composition.
The study appears to my opinion well performed and well discussed, and the data were analyzed with a robust statistical analysis.
I have only some comments during the revision process
- the introduction section appears very long, it could reduce the readability of the study. I suggest to reduce it
- “The required sample was estimated to be of 68 infants, 34 infants in each fortification method cohort”. Please describe more in detail
- I seem to understand that the final end point of growth parameters is the discharge. It is not well clarified in the abstract and in the methods section, or maybe I miss this information. The timing of discharge is the same between the two study Group? Please provide
Author Response
Authors performed an observational study to assess whether the implemented fortification method based on measured HM composition improves the energy and macronutrient intake and the quality of growth in infants born less than 33 weeks’ gestation, compared with HM fortification based on assumed HM composition.
The study appears to my opinion well performed and well discussed, and the data were analyzed with a robust statistical analysis.
I have only some comments during the revision process
Response: Thank you for your appreciation.
- the introduction section appears very long, it could reduce the readability of the study. I suggest to reduce it
Response: Thank you for the suggestion. Accordingly, the Introduction section was reduced eliminating non-essential information.
- “The required sample was estimated to be of 68 infants, 34 infants in each fortification method cohort”. Please describe more in detail.
Response: Thank you for the question and the opportunity to clarify. The sample size calculation is explained in the published study protocol which is cited in the manuscript. Nevertheless, to be more readily available, the sample size calculation has been included in the new Section 2.5 of the revised manuscript.
- I seem to understand that the final end point of growth parameters is the discharge. It is not well clarified in the abstract and in the methods section, or maybe I miss this information. The timing of discharge is the same between the two study Group? Please provide.
Response: Thank you for the question. You understood it correctly, the last point of anthropometry assessment was at discharge. As suggested, this is now clearly stated in revised Abstract and in Methods (Section 2.4). As stated in Supplementary Table S4, the timing of discharge did not differ significantly between Group 1 and 2, occurring at median of 36.2 and 37.0 postmenstrual weeks (p=0.082), respectively.

Round 2
Reviewer 1 Report
The authors have carefully addressed my concerns. However, while most things have been sufficiently edited, a few more details should be added in the Methods section:
- P5L203/204: What method was used to determine milk intake. This should be explained.
- P5, Anthropometry: Please add details regarding the equipment and protocol. Which scales were used? Did you do one measurement or multiple, e.g. for infant length, reports often states the measurement to the next full cm from an average of 3 measurements.
Generally, please consider reviewing your information from the reader’s perspective of not being part of your study. Thus, you should add all the information that should be provided to enable the unfamiliar reader to get the whole picture. Detailed methodological approaches are a vital part of any research paper.
Author Response
The authors have carefully addressed my concerns. However, while most things have been sufficiently edited, a few more details should be added in the Methods section.
Response: Thank you for the insightful questions and the opportunity to clarify.
- P5 L203/204: What method was used to determine milk intake. This should be explained.
Response: In the revised manuscript R2 the milk intake assessment is clearly explained: “The daily record of nutritional intake refers to foods administered in the last 24 hours. For each infant, the volumes of LH administered (MOM or DHM) were accurately recorded, considering the amount actually administered, taking into account possible interruptions and gastric residual volumes”
- P5, Anthropometry: Please add details regarding the equipment and protocol. Which scales were used? Did you do one measurement or multiple, e.g. for infant length, reports often states the measurement to the next full cm from an average of 3 measurements.
Response: In the revised manuscript R2 the details regarding the equipment and measurement technique used is explained in more detail.
Generally, please consider reviewing your information from the reader’s perspective of not being part of your study. Thus, you should add all the information that should be provided to enable the unfamiliar reader to get the whole picture. Detailed methodological approaches are a vital part of any research paper.

Reviewer 2 Report
all my questions have been resolved
congratulations
Author Response
All my questions have been resolved
Congratulations
Response: Thank you for the kind comment.